# Effect of 3D Spheroid Culturing on NF-κB Signaling Pathway and Neurogenic Potential in Human Amniotic Fluid Stem Cells

**DOI:** 10.3390/ijms24043584

**Published:** 2023-02-10

**Authors:** Giedrė Valiulienė, Aistė Zentelytė, Elizabet Beržanskytė, Rūta Navakauskienė

**Affiliations:** Department of Molecular Cell Biology, Institute of Biochemistry, Life Sciences Center, Vilnius University, Saulėtekio av. 7, LT-01257 Vilnius, Lithuania

**Keywords:** human amniotic fluid stem cells (hAFSCs), three-dimensional (3D) spheroid cultures, neural differentiation, NF-κB

## Abstract

Human amniotic fluid stem cells (hAFSCs) are known for their advantageous properties when compared to somatic stem cells from other sources. Recently hAFSCs have gained attention for their neurogenic potential and secretory profile. However, hAFSCs in three-dimensional (3D) cultures remain poorly investigated. Therefore, we aimed to evaluate cellular properties, neural differentiation, and gene and protein expression in 3D spheroid cultures of hAFSCs in comparison to traditional two-dimensional (2D) monolayer cultures. For this purpose, hAFSCs were obtained from amniotic fluid of healthy pregnancies and cultivated in vitro, either in 2D, or 3D under untreated or neuro-differentiated conditions. We observed upregulated expression of pluripotency genes *OCT4*, *NANOG*, and *MSI1* as well as augmentation in gene expression of NF-κB−TNFα pathway genes (*NFKB2*, *RELA* and *TNFR2*), associated miRNAs (miR103a-5p, miR199a-3p and miR223-3p), and NF-κB p65 protein levels in untreated hAFSC 3D cultures. Additionally, MS analysis of the 3D hAFSCs secretome revealed protein upregulation of IGFs signaling the cascade and downregulation of extracellular matrix proteins, whereas neural differentiation of hAFSC spheroids increased the expression of *SOX2,* miR223-3p, and MSI1. Summarizing, our study provides novel insights into how 3D culture affects neurogenic potential and signaling pathways of hAFSCs, especially NF-κB, although further studies are needed to elucidate the benefits of 3D cultures more thoroughly.

## 1. Introduction

Human amniotic fluid stem cells (hAFSCs) are known for their higher differentiation potential as they can differentiate into lineages of all three germ layers, they are also recognized for their low immunogenicity and minimally invasive isolation procedure, compared to mesenchymal stem cells (MSCs), obtained from other sources [1,2]. Furthermore, hAFSCs are less prone to senescence in comparison to bone marrow MSCs for example [3]. In addition, hAFSCs do not produce teratomas in vivo [4]. Although these features make hAFSCs an attractive tool for regenerative medicine purposes [5], especially for the treatment of neurodevelopmental and neurological disorders [6], knowledge about the characteristics of hAFSCs in three-dimensional (3D) spheroid cultures is still scarce.

Typically, MSCs have been researched in two-dimensional (2D) monolayer cultures. Only recently has 3D cell culturing emerged as a new tool that enables better representation of in vivo conditions, increasing cell-to-cell interactions [7]. The latest meta-analysis by Ezquerra et al. [8] revealed that MSCs from adipose tissue, bone marrow, and umbilical cord, when grown in 3D cultures, exhibit better potential of stemness, angiogenesis and differentiation in contrast to their 2D cell counterparts. In addition, advantages of in vivo settings were also observed, as MSCs spheroids demonstrated better immunomodulatory characteristics and more efficient engraftment, as well as higher MSCs survival. Despite the widely excepted benefits of 3D cell culturing, research regarding the three-dimensional hAFSC cultures is very limited and mainly involves analysis of cells grown on transplantable scaffolds [9,10,11]. Moreover, there is still no data on neurogenic properties of hAFSCs when grown in 3D cultures.

Therefore, in the present study, we aimed to explore the effect of hAFSC culturing in 3D spheroids vs. traditional monolayer conditions. We were also interested in whether the cell maintenance in 3D may increase the neurogenic potential of hAFSCs. Obtained results have revealed that culturing of hAFSCs in spheroids do indeed upregulate expression of stemness markers and certain neural progenitor markers. Augmentation in gene and protein expression of members in NF-κB−TNFα signaling pathway was also observed. Our findings indicate that hAFSCs in spheroid cultures exhibit some distinct, yet advantageous, molecular profiles compared to hAFSCs from monolayer cultures. For this reason, further more thorough investigations should proceed in order to clarify the potential of hAFSCs spheroids in the clinical settings.

## 2. Results

### 2.1. Evaluation of Morphological Characteristics and Expression of Cell Surface Markers in hAFSCs Grown in 2D and 3D Cultures

With the flow cytometry technique, we first evaluated the expression of cell surface markers in untreated hAFSCs grown in 2D cell cultures (Figure 1A). The results of this analysis confirmed that hAFSCs in conventional 2D cultures are highly positive for mesenchymal stem cell markers, such as CD44 (homing cell adhesion molecule; up to 95% positive cells), CD56 (neural cell adhesion molecule 1, NCAM1; up to 78% positive cells), CD90 (Thy-1, GPI-linked glycoprotein; up to 95% positive cells), CD105 (Endoglin; up to 89% positive cells), CD146 (melanoma cell adhesion molecule; up to 82% positive cells), CD166 (activated leukocyte cell adhesion molecule; up to 95% positive cells), HLA-ABC (MHC class I antigen; up to 96% positive cells) and SSEA-4 (up to 98% positive cells). No or only negligible expression of CD31 (platelet endothelial cell adhesion molecule), CD34 (hematopoietic progenitor cell antigen) and HLA-DR (MHC class II antigen) was detected.

Further, we characterized hAFSCs morphology, when cells were grown in monolayer and 3D spheroid cultures (Figure 1B). hAFSCs were either untreated (control) or neuro-induced. For neurogenic differentiation induction, cells were pretreated for 1 day with preinduction media supplemented with 20 ng/mL FGF and 20 ng/mL EGF, and they were then further treated for 5 days with a chemical cocktail consisting of 50 ng/mL BDNF + 100 ng/mL NGF + 5 mM KCl + 2 μM RA. For spheroid formation, 10,000 cells/per well (Nunclon Sphera 96-well plate) were used. Untreated and neuro-induced hAFSCs spheroids were cultivated for 7 days in total.

As is evident from the light microscopic examination, the control (untreated) hAFSCs in 2D cultures exhibited typical mesenchymal stem cell morphology, whereas neural induction in 2D cultures upregulated hAFSCs into a more neural-like elongated appearance. However, no differences in 3D spheroid morphology were observed in treated vs. untreated cultures, as both types of spheroids measured approx. 285 µm in diameter. We have also examined the characteristics of hAFSCs migrated from 3D spheroids (after 7 days in culture, spheroids were reattached from the 96-well plates, seeded to cell culture dishes, and left for 5 days for cells to migrate). Evidently, migrated monolayer hAFSCs, either from untreated or neuro-induced spheroids, shared similar morphological appearance. Only a minority of cells from neuro-induced hAFSCs spheroids exhibited typical neural-like cell morphology (denoted with arrows in Figure 1B).

### 2.2. Gene and miRNA Expression Patterns in Control and Neuro-Differentiated hAFSCs Grown in 2D and 3D Cultures

Gene expression levels of neuro-differentiated hAFSCs, grown either in 2D or 3D spheroid cultures, were evaluated after 24h in preinduction media (20 ng/mL FGF and 20 ng/mL EGF) and 5 days in induction media (neurogenic differentiation of hAFSCs was induced using chemical cocktail consisting of 50 ng/mL BDNF + 100 ng/mL NGF + 5 mM KCl + 2 μM RA). Results were compared to untreated hAFSCs, grown in 2D and 3D conditions. RT-qPCR analysis (Figure 2A,B) revealed that culturing of untreated hAFSCs in three-dimensional spheroids, in comparison to growing in traditional 2D cultures, upregulated expression of stemness markers such as *SOX2* (up to 72-fold, ns), *OCT4* (up to 11-fold, ns), *NANOG* (up to 14-fold, *p* < 0.05), *LIN28A* (up to 4.7-fold, *p* < 0.05) and *MSI1* (up to 329-fold, ns). In addition, gene expression of neural lineage-associated genes, such as *NTRK1* (Neurotrophic receptor tyrosine kinase 1; receptor of NGF) was increased approx. 9-fold (*p* < 0.05), whereas expression of pluripotent neurotrophic factor *CNTF* (Ciliary neurotrophic factor) and transcription factor *NEUROD1* (Neurogenic differentiation 1) was upregulated 20- and 51-fold in untreated hAFSCs 3D vs. 2D cultures, respectively (*p* < 0.01). Interestingly, hAFSCs spheroids that were cultured in neurogenic induction media were not characterized by higher neuronal marker expression in comparison to 3D untreated hAFSCs (Figure 2B), except the *SOX2*, which expression was augmented more strongly in neuro-induced than untreated hAFSCs 3D cultures (up to 5-fold, ns). Gene expression of *SOX2* was also more significantly pronounced (more than 30-fold; *p* < 0.05) in neuro-induced 3D cell cultures than in 2D differentiated cells. Surprisingly, gene expression of proto-oncogenic factor *MYC*, which is also known for its role in neuronal differentiation [12], was significantly higher in 2D than in 3D neuro-induced cells (more than 5-fold, *p* < 0.01). To sum up, it should be emphasized that the 5-day-long treatment of hAFSCs in 3D cultures with a combination of BDNF, NGF, RA and KCl upregulated stemness and neural differentiation-associated gene expression in comparison to treated 2D cultures; however, such a modulation was mostly dependent on the three-dimensional architecture itself.

Subsequent miRNA analysis (Figure 3A) also revealed similar results, as untreated hAFSCs in 3D cultures had the highest expression of stemness and neural processes-associated miRNAs, such as miR103a-5p (*p* < 0.05), miR199a-3p (*p* < 0.05), as well as miR146a-5p (*p* < 0.05). However, miR10-5p, which is known to directly target and inhibit PTEN, is a factor that is crucial for stem cell maintenance [13,14], was the most strongly upregulated in 2D neuro-induced hAFSCs cultures (up to 4-fold in comparison to untreated 2D control, *p* < 0.01), whereas expression of miR223-3p, which regulates the differentiation of immature neurons [15], was significantly augmented in 3D neuro-induced cells (more than 304-fold in comparison to untreated 2D control, *p* < 0.01; and up to 10-fold in comparison to untreated 3D hAFSCs, *p* < 0.05).

As upregulated miR199a-3p, miR146a-5p and miR223-3p are widely accepted to be associated with NF-κB (Nuclear factor κB) signaling pathway modulation [16,17,18], we further examined the gene expression of NF-κB signaling pathway-related genes (Figure 3B). The obtained results in 3D untreated hAFSCs spheroid cultures indicated the significant upregulation of genes *NFKB2* (up to 5-fold compared to 2D control; *p* < 0.01) and *TNFR2* (up to 24-fold compared to 2D control; *p* < 0.05), which stands for subunit 2 of transcription factor NF-κB and Tumor necrosis factor receptor 2, respectively. In addition, in 3D cultures of neuro-induced hAFSCs spheroids, the augmentation of *REL* gene expression, the transcription factor from the NF-κB family, was observed (increased up to 6.5-fold compared to 2D control; *p* < 0.05). All these results indicate that hAFSC culturing in 3D spheroids do modulate gene and miRNA expression that is associated with the processes of stemness, neurogenesis and regulation of cell survival.

### 2.3. Modulation of Protein Expression in Control and Neuro-Differentiated hAFSCs Grown in 2D and 3D Cultures

Next, we evaluated and compared protein expression of untreated and neuro-differentiated hAFSCs in 2D and 3D cultures. First, hAFSCs migrated from spheroids (after 5 days of reattachment and seeding to cell culture dishes) were examined with flow cytometry (Figure 4A,B). Results demonstrated that neural induction with 50 ng/mL BDNF + 100 ng/mL NGF + 5 mM KCl + 2 μM RA in 3D spheroid cultures upregulated the quantity of hAFSCs positive for Lin28a and MSI1 (only a negligible increase in Nestin protein expression was detected). The most significant increase (*p* < 0.05) was observed in MSI1+ cells (from 47% in untreated 3D cultures to up to 68% in neuro-differentiated 3D cultures). The flow cytometry analysis results were confirmed by immunofluorescent evaluation of LIN28a, MSI1, and Nestin expression in control and neuro-induced hAFSCs migrated from 3D spheroids (Figure 4C).

Augmentation in expression of pro-apoptotic proteins MYC and BAX in neuro-differentiated hAFSCs (2D cultures and spheroids) was confirmed with Western blot and immunofluorescence techniques (Figure 5A,B). Surprisingly, MYC expression was detected not only in nucleus but also in cytoplasm of 2D neuro-induced hAFSCs (see Appendix A). In addition, the results of protein analysis coincided with the data of gene expression (Figure 2A), as MYC and BAX protein levels were higher in 2D rather than in 3D neuro-induced cells. Contrary, expression of anti-apoptotic MCL1 protein increased at a comparable degree in both 2D and 3D neuro-induced hAFSCs, as well as in untreated 3D spheroids. Similarly to the mRNA levels of TUBB3 (Figure 2B), no differences were detected in the protein quantities of TUBB3, either in untreated or induced to neural differentiation hAFSCs, when grown in 2D or 3D cell cultures. Although no significant differences in *RELA* gene expression was detected either in untreated 3D, or neuro-induced 2D and 3D hAFSCs cultures, Western blot analysis revealed that the levels of NFκB p65 subunit were upregulated in neuro-differentiated 2D (up to 1.45-fold, n. s.) and untreated 3D hAFSCs (up to 1.6-fold, *p* = 0.05). These results indicate that neural differentiation and three-dimensional spheroid formation in hAFSCs are accompanied by the modulation of pro-survival and apoptotic pathways.

### 2.4. Examination of hAFSCs Secretion Profile in 2D and 3D Cultures

We used mass spectrometry technique for analysis of proteins that were secreted by the hAFSCs, cultivated either in 2D, or 3D conditions. A total of 157 cell-specific hAFSCs’ secreted proteins were detected in samples (see Appendix A).

We further analyzed separately the proteins that were upregulated (33 in total) or downregulated (32 in total) in a 3D hAFSCs secretome in comparison to a 2D-grown hAFSCs secretome, and we depicted them using STRING protein–protein interaction networks (Figure 6A,B). Proteins were also classified accordingly to the pathways they represent in the Reactome database. The 25 most relevant pathways from Reactome the database sorted by *p*-value were denoted in Appendix A.

Results revealed high upregulation (up to 18-fold) of Alpha-2-macroglobulin (A2M) in the secretome of 3D hAFSCs. According to Liu et al. [19], this large glycoprotein may have a protective effect on mesenchymal stem cells. In addition, mass spectrometry data showed that certain proteins belonging to the pathway of Insulin-like growth factor (IGF) transport regulation and uptake by Insulin-like growth factor binding proteins (IGFBPs) (e.g., Ceruloplasmin (CP), Complement C3 (C3) and Apolipoprotein A-II (APOA2)) were more intensely secreted from hAFSCs grown in 3D rather than in 2D conditions (Figure 6A). For example, Complement C3 was upregulated up to 15-fold (Table 1). In addition, an increase in proteins associated with complement activation (Immunoglobulin kappa constant (IGKC) and Carboxypeptidase N subunit 2 (CPN2); up to 3.1-fold) and the transport of fatty acids (Apolipoprotein D (APOD); up to 1.4-fold) were observed. Furthermore, a decrease in TGFBI (Transforming growth factor beta-induced protein) in the 3D cell secretome (more than 409-fold) was registered (Figure 6B, Table 1), which negatively correlated with the intracellular levels of RelA protein (Figure 5A). Regarding data in the literature, both upregulation of IGF pathway proteins and downregulation of TGFBI protein level in the 3D hAFSCs secretome may be associated with NF-κB pathway activation due to 3D cultivation conditions [20,21].

A noticeable decline was observed in collagen secretion (COL5A1, COL1A1; reduction up to 36 and 28-fold, respectively), as well as decrease in collagen processing enzymes (SERPINE1, up to 484-fold, SPARC, up to 195-fold and TIMP1, up to 83-fold) and other components of extracellular matrix (Fibronectin, up to 83-fold and Galectin-1, up to 17-fold). All these events obviously correspond to complex remodulation of extracellular matrix in 3D hAFSCs cultures.

## 3. Discussion

Culturing of mesenchymal stem cells in 2D conditions (monolayers) provides a fast and convenient method of cell expansion. However, 3D cultures are widely accepted as a more preferable technology because 3D cultures (spheroids) do resemble the physiological conditions more accurately. Furthermore, MSCs in 3D spheroids were shown to maintain their properties of stemness for longer periods in comparison to MSCs in monolayers [22]. In this study, we aimed to investigate whether 3D spheroids change the stemness properties and neurogenic/neurotrophic potential of hAFSCs.

The results of our study revealed that hAFSCs in spheroids exhibit a higher expression of pluripotency genes *SOX2*, *OCT4*, *NANOG*, *LIN28A* and *MSI1* (Figure 2A). This is in agreement with the data of other authors, who demonstrated that MSCs form other sources (endometrium), when grown in spheroids, do display upregulation of pluripotency genes as well [23]. These findings also coincide with the results of Niibe et al. [24], who demonstrated that culturing of human bone marrow-derived mesenchymal stem cells (hBM-MSCs) in 3D spheroids do increase cell surface expression of certain stemness markers (CD106 and CD271) in comparison to 2D-grown hBM-MSCs.

Although no morphological differences were detected between untreated and neuro-induced hAFSCs spheroids, as well as migrated cells from these spheroids, a distinction could be made regarding gene, miRNA, and protein expression. For example, the gene expression of *SOX2*, transcription factor that is key for pluripotency, as well as neurogenic lineage [25], was also significantly augmented in the neuro-induced hAFSCs spheroids compared to 2D untreated and neuro-differentiated cells. Additionally, the gene and protein expression of neural marker Nestin, RNA binding protein Musashi 1, and transcription factor LIN28a was slightly higher in neuro-induced spheroids in comparison to untreated spheroids. As MSI1 is well-known for stemness promotion in cancer cells as well as mesenchymal stem cells [26,27], upregulation of *MSI1* in hAFSCs 3D spheroid cultures could indicate increased stemness potential when compared to 2D cultures. In addition, MSI1 is one of the main factors that control neural lineage differentiation [28,29]. Therefore, not surprisingly, higher MSI1 protein expression was detected in neuro-induced hAFSCs spheroids than in control (untreated) spheroids. Similarly, higher protein expression of MYC was found in the neuro-induced hAFSCs spheroids in comparison to untreated spheroids. This could be also explained by the fact that MYC is crucial not only for the maintenance of stem cells, but it plays a significant role in the neural differentiation as well [30]. Surprisingly, in our study MYC demonstrated cytoplasmic as well as nuclear localization in neuro-induced hAFSCs from 2D cultures. As summarized by Conacci-Sorrell et al. [31], there have been multiple reports of cytoplasmically localized MYC, predominantly in differentiated cells and cytoplasmic forms of MYC therefore may be involved in neural differentiation processes in hAFSCs as well. Of course, further studies are necessary to clarify the hypothesis.

However, results of our study indicated that neural differentiation of hAFSCs induced expression of glial lineage markers, rather than neuronal ones. For example, augmented expression of miR146a-5p and miR223-3p is assumed to be associated with the glial phenotype [32]. Upregulation of gene expression of Glial fibrillary acidic protein (*GFAP*), as well as Aldehyde dehydrogenase 1 family member L1 (*ALDH1L1*) and neurotrophic factor *CNTF* in the neuro-induced hAFSCs spheroids may also be related to the astrocytic phenotype [33]. Nevertheless, it should be stressed out that the highest level of miR146a-5p was detected in untreated 3D spheroids, rather than in the neuro-induced 3D cultures. Untreated hAFSCs spheroids were also the most characteristic for expression of other miRNAs, such as miR103a-5p, miR199a-3p and miR223-3p. As aforementioned miRNAs are widely accepted to be involved NF-κB signaling pathway [17,18,34,35], we tested the gene and protein expression of this molecular network further. In 3D untreated hAFSCs spheroid cultures, the upregulation of *NFKB2* and *TNFR2* was evident. In addition, in untreated hAFSCs spheroids, the levels of NF-κB p65 subunit (coded by *RELA* gene) were the most strongly upregulated, compared to untreated 2D control or neuro-differentiated 2D and 3D cultures, although the changes were not statistically significant. It was recently shown by Yu et al. [36] that the NF-κB p65 subunit promotes proliferation but inhibits osteogenic and chondrogenic differentiation of MSCs. Therefore, an increase in the NF-κB p65 protein level in untreated 3D cultures and downregulation upon 3D neuro-differentiation may indicate similar processes in hAFSCs as well. It is worth mentioning that in 3D hAFSCs cultures, both in untreated and neuro-induced cells, a positive correlation between miR103a-5p expression and NF-κB p65 protein levels was detected. As the Lu et al. [37] study showed, miR103a-5p elicits its biological functions via SNRK (SNF-related serine/threonine-protein kinase) inhibition and subsequent activation of NFκB p65.

In addition, the detected changes in the protein secretion profile of hAFSCs, when grown in 3D spheroids, may also be associated with the modulation of the NF-κB signaling pathway. For example, the upregulation of IGF signaling pathway proteins may possibly stimulate NF-kB growth-promoting effects [38]. In accordance to this, human mesenchymal stem cells were previously shown to increase the secretion of growth factors (VEGF, FGF2, IGF1 and HGF) via NF-κB pathway under hypoxic conditions [39]. Contrarily, suppression of TGFBI protein secretion from 3D-grown hAFSCs may also indicate the increase in NF-κB signaling activity, and as such, antagonism was demonstrated earlier by other authors [20]. Consequently, the results of our study revealed that a 3D spheroid formation in hAFSCs is associated with NF-κB pathway activation. As previously demonstrated by others, the activation of the NF-κB pathway is linked to the 3D organization of human breast cancer cells [40]. In addition, NF-κB signaling was also shown to enhance differentiation potential and stemness capacities of human dental pulp-derived MSCs in 3D spheroid cultures [41].

As MSCs cultured in 3D spheroids may be predisposed to apoptosis induction under certain stressors [42], we tested the effect of changing culture conditions (3D vs. 2D, and neuro-induced vs. untreated) on hAFSCs gene and protein expression of certain apoptosis-associated factors, such as the aforementioned c-MYC, and also MCL1, BAK, and BAX. The results of our analysis revealed that maintaining untreated hAFSCs in 3D cultures for 7 days does not significantly increase either gene or protein expression of apoptosis regulator BAX. However, quite notable upregulation of BAX protein expression was evident upon neurogenic differentiation in 2D and 3D cultures Additionally, the gene expression of pro-apoptotic factor *BAK1* was significantly downregulated in 2D and 3D neuro-differentiated hAFSCs, in comparison to untreated controls. Recently, it was demonstrated that in addition to the canonical activities of BAX and BAK in apoptosis, these factors have non-canonical functions in mitochondrial dynamics as well as morphology regulation, and consequently, they are crucial for both neural stem cells and whole-brain development [43]. Therefore, in this context, BAX- and *BAK1*-level modulation may be associated with neural differentiation processes rather than with apoptosis induction, as an increase in anti-apoptotic MCL1 protein levels was observed in all culture conditions compared to 2D control.

## 4. Materials and Methods

### 4.1. Isolation, Cultivation and Spheroid Formation of Human Amniotic Fluid Stem Cells

Human amniotic fluid samples were obtained by amniocentesis from mid-second-trimester pregnancies (16–17 weeks of gestation; patient age: 39 ± 2.1 years; n = 3) from healthy women who needed prenatal diagnostics, but no genetic abnormalities or aneuploidies were detected during clinical diagnostics. Protocols approved by the Vilnius Regional Biomedical Research Ethics Committee (No. 158200-18/7-1049-550), all patients involved in this study signed a written consent.

AFSCs were isolated using a two-stage protocol as previously described [44] and cultivated in complete growth medium consisting of basal DMEM (4.5 g/L glucose) media, 10% FBS, 100 U/mL penicillin, and 100 mg/mL streptomycin (Gibco, ThermoFisher Scientific, Waltham, MA, USA). The 3D cultures (spheroids) were formed using Nunclon^TM^ Sphera^TM^ (ThermoFisher Scientific, Waltham, MA, USA) 96-well plates, AFSCs were seeded at 1 × 10^4^-cell/well density in complete growth medium and cultivated for 7 days.

### 4.2. Differentiation Assay

Neurogenic differentiation in monolayer culture was induced when cell confluency reached 60%, and 3D spheroids were induced to differentiate 24 h after seeding. To induce differentiation preinduction media consisting of DMEM (1 g/L glucose), 10% heat-inactivated FBS, 100 U/mL penicillin, and 100 mg/mL streptomycin (Gibco, ThermoFisher Scientific, Waltham, MA, USA), 20 ng/mL FGF, and 20 ng/mL EGF (PeproTech, London, UK) was applied for 24 h. After pre-induction step differentiation media consisting of BrainPhys™, 1% NeuroCult™ (STEMCELL Technologies, Vancouver, BC, Canada), 100 U/mL penicillin, and 100 mg/mL streptomycin, 50 ng/mL BDNF, 100 ng/mL NGF, (PeproTech, London, UK) 5 mM KCl, 2 μM RA (Sigma-Aldrich, St. Louis, MO, USA). Differentiation was carried out for 5 days.

### 4.3. Flow Cytometry Analysis

AFSCs were characterized by the expression of surface markers. Collected AFSCs from monolayer cultures were washed twice using PBS with 1% bovine serum albumin (BSA) (Genaxxon bioscience, Ulm, Germany) and a total 6 × 10^4^ cells were resuspended in the PBS/BSA solution. AFSCs were incubated with antibodies indicated in Appendix A in the dark, at 4 °C, for 30 min. After incubation, AFSCs were washed and analyzed using Millipore Guava^®^ easyCyte 8HT flow cytometer with InCyte 2.2.2 software. Ten thousand events were collected for each sample.

To measure protein expression in undifferentiated and differentiated 3D cultures, spheroids were transferred to adherent culture plates and left to migrate for 5 days. Migrated cells were collected by trypsinization, washed with PBS/BSA solution, fixed using 2% PFA (Sigma-Aldrich, St. Louis, MO, USA) and then permeabilized with PBS/BSA/0.1% Triton X-100 (Honeywell, Charlotte, NC, USA). Resuspended in PBS/BSA/0.1% Triton X-100 solution AFSCs were labelled with primary antibodies against, LIN28a, MSI1, Nestin and then with secondary Alexa Fluor^®^ 488 antibodies (Appendix A). Both incubation steps were carried out in the dark, at 4 °C, for 30 min. After incubation, AFSCs were washed and analyzed using Millipore Guava^®^ easyCyte 8HT flow cytometer with InCyte 2.2.2 software. Ten thousand events were collected for each sample.

### 4.4. RNA Isolation and RT-qPCR

Total RNA from both monolayer and 3D cultures of AFSCs was isolated using TRIzol^®^ reagent (Invitrogen, Carlsbad, CA, USA) as recommended by the manufacturer. Isolated RNA was further purified with RNase-free DNase I Kit (Thermo Scientific, Vilnius, Lithuania) and cDNA was synthesized by using SensiFAST^TM^ cDNA Synthesis Kit (Bioline, London, UK). RT-qPCR was performed with SensiFAST^TM^ SYBR^®^ No-ROX kit (Bioline, London, UK) on the Rotor-Gene^TM^ 6000 thermocycler with Rotor-Gene 6000 series software (Corbett Life Science, QIAGEN, Hilden, Germany). mRNA levels were normalized by GAPDH and RPL13A (geometric mean), and relative gene expression was calculated using ∆∆Ct method and compared to undifferentiated control. The list of used primers (Metabion International AG, Planegg/Steinkirchen, Germany) is provided in Appendix A.

### 4.5. Evaluation of Micro RNAs

MicroRNAs (miR) expression levels were evaluated using TaqMan Advanced miRNR Assays (Thermo Fisher Scientific, Waltham, MA, USA) according to manufacturer’s instructions. The following microRNAs were analyzed: miR-16-5p, miR-199a-3p, miR-93-5p, miR-10a-5p, miR-210-3p, miR-155-5p, miR-103-3p, miR-146a-5p, miR-223-3p. The expression of selected miRs were determined by RT-qPCR on the Rotor-Gene^TM^ 6000 thermocycler with Rotor-Gene 6000 series software (Corbett Life Science, QIAGEN, Hilden, Germany). miR-423-5p was used as a reference miR to normalize all experimental data, the relative expression was calculated using ∆∆Ct method and compared to undifferentiated control.

### 4.6. Immunofluorescence Analysis

For immunofluorescence, hAFSCs were seeded as monolayer culture in Lab-Tek Chamber slides (Thermo Fisher Scientific, Waltham, MA, USA), or 3D culture was formed using the Nunclon^TM^ Sphera^TM^ (ThermoFisher Scientific, Waltham, MA, USA) 96-well plates as described previously, the cells were cultivated as control or differentiated towards neurogenic lineage. Additionally, several spheroids were transferred to Lab-Tek Chamber slides, and cells were left to migrate for 5 days. hAFSCs in 2D or 3D culture were fixed with 4% PFA (Sigma-Aldrich, St. Louis, MO, USA) solution for 15 min, at RT, washed with PBS, and then permeabilized using 10% Triton X-100/PBS (Honeywell, Charlotte, NC, USA) for 20 min, at RT, washed again with PBS, and then blocked with 1% BSA/10% goat serum/PBS for 30 min, at 37 °C. For detection of MSI1, hAFSCs were incubated with primary rabbit antibodies against MSI1 (1:200) (Novus Biologicals, Abingdon, UK) for 1 h, at 37 °C, followed by incubation with secondary goat anti-rabbit IgG (H + L) Highly Cross-Adsorbed, Alexa Fluor^®^ 594 antibodies (1:400) (Invitrogen, Thermo Fisher Scientific, Waltham, MA, USA) for 1 h, at 37 °C. For detection of LIN28a, Nestin, and MYC, cells were incubated with primary mouse antibodies against LIN28a (1:200) and Nestin (1:100) or rabbit antibodies against MYC (1:150) (Novus Biologicals, Abingdon, UK) for 1 h, at 37 °C, followed by incubation with secondary goat anti-rabbit or goat anti-mouse IgG (H + L) Highly Cross-Adsorbed, Alexa Fluor^®^ 488 antibodies (1:400) (Invitrogen, Thermo Fisher Scientific, Waltham, MA, USA) for 1 h, at 37 °C. After each incubation cells were washed four times with 1% BSA/PBS. Nuclei were stained with 300 nM DAPI solution (Invitrogen, Thermo Fisher Scientific, Waltham, MA, USA) for 10 min, at RT and slides were mounted with Dako Fluorescent Mounting Medium (Agilent Technologies, Santa Clara, CA, USA). Labeled cells were analyzed using Zeiss Axio Observer (Zeiss, Oberkochen, Germany) fluorescent microscope, 63x objective magnification with immersion oil and Zen BLUE software.

### 4.7. Protein Isolation and Western Blot Analysis

Proteins were isolated using lysis solution (62.5 mM Tris, pH 6.8, 100 mM DTT and 2% SDS, 10% glycerol, traces of bromphenol blue) and benzonase (Pure Grade, Merck, Darmstadt, Germany) as previously described [45]. Isolated proteins were separated on a 7% to 15% polyacrylamide gradient SDS-PAGE gel and then transferred to a PVDF membrane (Immobilon P; Millipore, Billerica, MA, USA). Membranes were first incubated with primary antibodies against MYC (Novus Biologicals, Abingdon, UK), TUBB3 (Abcam, Cambridge, UK), NF-κB p65 (Cell Signaling Technology, Danvers, MA, USA), MCL1 (Proteintech, Rosemont, IL, USA), BAX (Cell Signaling Technology, Danvers, MA, USA), ACTB (Abcam, Cambridge, UK) and then with goat anti-rabbit or rabbit anti-goat horseradish peroxidase-linked secondary antibodies (Agilent Dako, Santa Clara, CA, USA). Protein detection was performed with Clarity Western ECL Substrate (BioRad, Hercules, CA, USA) according to manufacturer’s instructions on ChemiDoc XRS+ System (BioRad, Hercules, CA, USA). Protein band intensity was evaluated densitometrically using ImageJ 1.45S software, and final values were calculated by dividing the test proteins by ACTB and then normalizing obtained values by the control protein.

### 4.8. Secretome Preparation for Mass Spectrometry

The 2D and 3D cultures of AFSCs were cultivated in complete growth medium till confluent (monolayer culture) or for 7 days (spheroids), then cell cultures were washed with PBS and NutriStem^®^ (Biological Industries, Kibbutz Beit-Haemek, Israel) media was applied. Conditioned media was collected after 3 days and centrifuged at 2000× *g* for 10 min to remove cell debris. The supernatant was collected and filtered through a 0.22 µm filter and the samples were further cleaned and concentrated using ProteoMiner™ Sequential Elution Large-Capacity kit (Bio-Rad Laboratories, Hercules, CA, USA) according to the manufacturer’s instructions. The concentration of isolated proteins was measured with Pierce Detergent Compatible Bradford Assay Kit (Thermo Fisher Scientific, Waltham, MA, USA) according to the manufacturer’s instructions, and equal amounts of proteins were taken for further analysis. Trypsin digestion of proteins was performed according to a modified FASP protocol, as described by Wiśniewski and colleagues [46]. Briefly, samples were washed with buffer containing 8 M urea, and proteins were alkylated with 50 mM iodoacetamide (GE Healthcare Life Sciences, Marlborough, MA, USA). Proteins were washed twice with urea and twice with 50 mM NH_4_HCO_3_ and digested overnight with TPCK Trypsin 20233 (Thermo Scientific, Vilnius, Lithuania). After digestion, peptides were recovered by centrifugation at 14,000× *g* for 10 min and then two additional washes using 20% CH_3_CN. The peptides were acidified with 10% CF_3_COOH and lyophilized in vacuum centrifuge. The lyophilized peptides were redissolved in 0.1% formic acid and then analyzed by mass spectrometry.

### 4.9. LC-MS-Based Protein Identification

Liquid chromatography (LC) was performed in a Waters Acquity ultra performance LC system (Waters Corporation, Wilmslow, UK). Peptides were separated on an ACQUITY UPLC HSS T3 250 mm analytical column. Data were acquired by using Synapt G2 mass spectrometer (MS) and Masslynx 4.1 software (Waters Corporation, Wilmslow, UK) in positive ion mode using data-independent acquisition (UDMSE). Raw data were lock mass-corrected using the doubly charged ion of [Glu1]-fibrinopeptide B (m/z 785.8426; [M + 2H]2+). Raw data files were processed and searched using ProteinLynx Global SERVER (PLGS) version 3.0.1 (Waters Corporation, Wilmslow, UK). Data were analyzed using trypsin as the cleavage protease; one missed cleavage was allowed, and fixed modification was set to carbamidomethylation of cysteines; variable modification was set to oxidation of methionine. Minimum identification criteria included 1 fragment ion per peptide, 3 fragment ions and 1 peptide per protein. The following parameters were used to generate peak lists: (i) low energy threshold was set to 150 counts; (ii) elevated energy threshold was set to 50 counts; (iii) intensity threshold was set to 750 counts. UniprotKB/SwissProt human databases (12 July 2021) were used. Protein quantification was calculated using the ISOQuant software. The obtained results of mass spectrometry analysis were also analyzed with the online tool STRING (String consortium, https://string-db.org, accessed on 13 July 2021) and Reactome pathway database (https://reactome.org/, accessed on 13 July 2021).

### 4.10. Statistical Analysis

All experiments were performed in triplicate, and data were expressed as the mean ± SD. Statistical analysis was conducted using Student’s *t*-test and one-way ANOVA with Tukey’s post hoc test in GraphPad Prism software.

## 5. Conclusions

When researching cellular properties becomes a priority, culturing conditions have to be taken into account, as 3D cultures are said to be a closer alternative to native conditions than the classical monolayer technique. In our study, we analyzed hAFSCs grown in untreated or neuro-differentiated 2D and 3D conditions and demonstrated that the cultivation of hAFSCs in 3D spheroids increases the gene expression of core pluripotency markers and modulates the NF-κB signaling pathway, as seen by the changed expression of genes, miRNAs, and proteins associated with aforementioned cellular properties and processes. Upregulation of NF-κB signaling pathway in hAFSCs 3D spheroid cultures may also be associated with the modulation of hAFSCs stemness potential itself, and should be thoroughly investigated further in the future studies, since establishing the exact driving forces behind these changes would be valuable for hAFSC applicability in clinical settings.

## Figures and Tables

**Figure 1 ijms-24-03584-f001:**
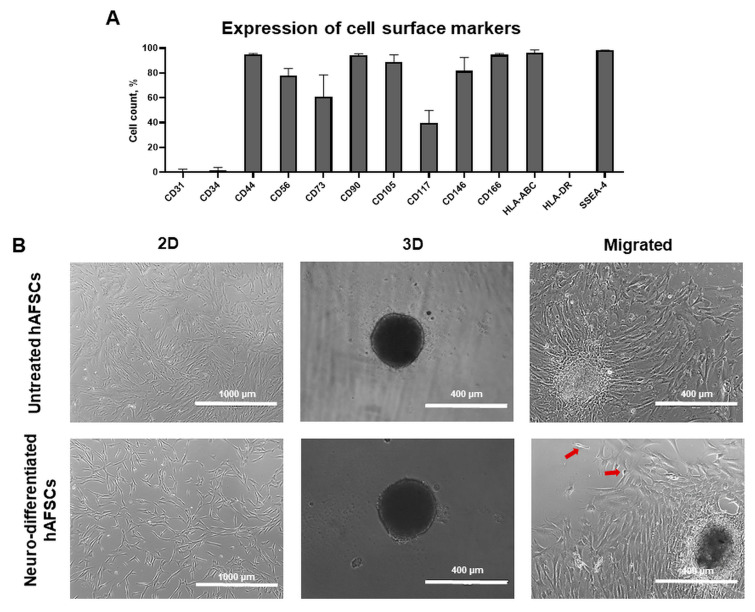
Morphological and cell surface marker characterization of hAFSCs grown in 2D and 3D cell cultures. (**A**) Flow cytometric cell surface marker expression analysis of CD31, CD34, CD44, CD56, CD73, CD90, CD105, CD117, CD146, CD166, HLA-ABC, and HLA-DR in control (untreated) hAFSCs, cultivated in 2D cell cultures. Data are shown as percentage (n = 3), and values are indicated as mean ± SD. (**B**) Morphology of control (untreated) and neuro-differentiated hAFSCs, grown in 2D and 3D (spheroid) cell cultures, and hAFSCs cells migrated from spheroids (representative images; scale bars: 1000 μm, 400 μm and 400 μm, respectively). Neurogenic differentiation was initiated using chemical cocktail consisting of 50 ng/mL BDNF + 100 ng/mL NGF + 5 mM KCl + 2 μM RA (representative image illustrating hAFSCs morphology in 2D after 24 h post differentiation induction). For spheroid formation, 10,000 hAFSCs were seeded to Nunclon Sphera 96-well plates. Spheroids were cultivated for 7 days. For hAFSCs migration from 3D cultures, spheroids, after 7 days of cultivation in Nunclon Sphera 96-well plates, were transferred to standard cell culture dishes and were left for 5 days to migrate. Red arrows denote cells that exhibited typical neural-like morphology.

**Figure 2 ijms-24-03584-f002:**
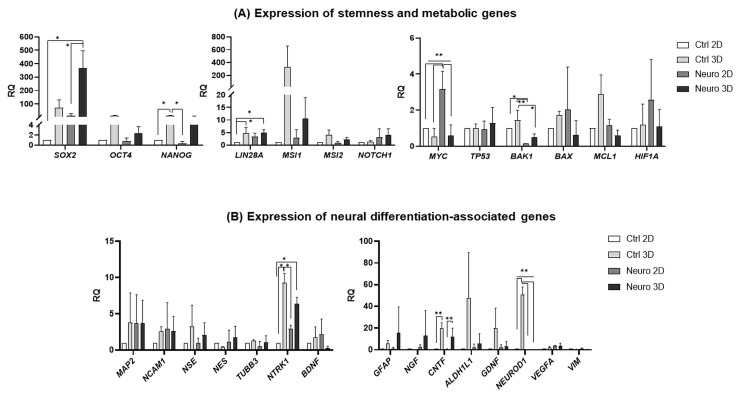
Gene expression profile of stemness, metabolic genes, and neuronal markers in control and induced to neural differentiation hAFSCs, grown in 2D and 3D cell cultures. RT-qPCR analysis of hAFSCs (**A**) stemness-associated genes *SOX2*, *OCT4*, *NANOG*, *LIN28A*, *MSI1*, *MSI2*, and *NOTCH1* and cell survival-associated/metabolic genes *MYC*, *TP53*, *BAK-1*, *BAX*, *MCL1*, *HIF1A*, (**B**) as well as neural differentiation crucial genes *MAP2*, *NCAM1*, *NSE*, *NES*, *TUBB3*, *NTRK1*, *BDNF*, *GFAP*, *NGF*, *CNTF*, *ALDH1L1*, *GDNF*, *NEUROD1*, *VEGF*, and *VIM*. Gene expression analysis was performed using control hAFSCs (not treated, “Ctrl”) and neuronal differentiation-induced hAFSCs (treated for 5 days with 50 ng/mL BDNF + 100 ng/mL NGF + 5 mM KCl + 2 μM RA; denoted as “Neuro”). Control and neuro-induced hAFSCs were cultivated either in 2D or 3D cell cultures. RT-qPCR data are represented as relative fold change over 2D undifferentiated control, normalized for the housekeeping genes GAPDH and RPL13A; values are indicated as mean ± SD (n = 3). Note: * denotes significant difference with *p* < 0.05, ** denotes significant difference with *p* < 0.01, as evaluated using one-way repeated-measures ANOVA with Tukey‘s post hoc test.

**Figure 3 ijms-24-03584-f003:**
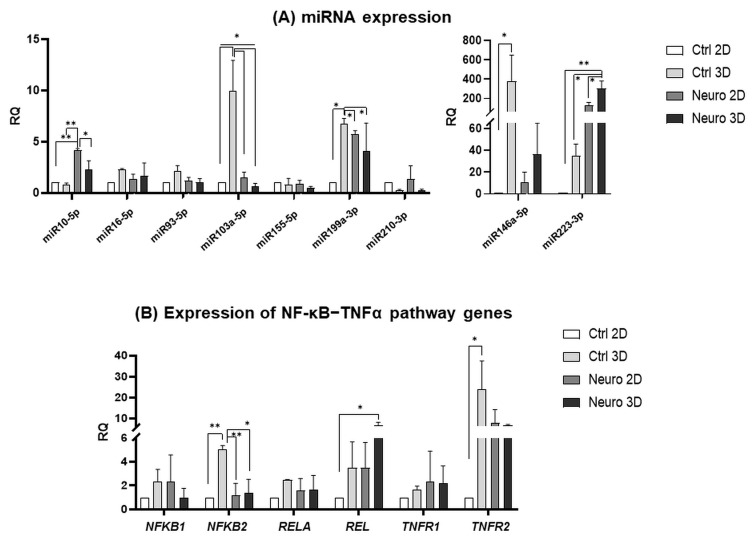
miRNA and gene expression profile of NF-κB–TNFα pathway genes in control (not treated, “Ctrl”) hAFSCs and hAFSCs induced to neural differentiation (“Neuro”; treated for 5 days with 50 ng/mL BDNF + 100 ng/mL NGF + 5 mM KCl + 2 μM RA; denoted as “Neuro”), grown in 2D and 3D cell cultures. (**A**) miRNA expression analysis of control hAFSCs and neuro-induced hAFSCs. Data are represented as relative fold change over 2D undifferentiated control, normalized for the endogenous control hsa-miR-423-5p; values are indicated as mean ± SD (n = 3). (**B**) Gene expression analysis of hAFSCs NF-κB–TNFα pathway-associated genes (*NFKB1*, *NFKB2*, *RELA*, *REL*, *TNFR1* and *TNFR2*). Gene expression analysis was performed using control hAFSCs and neuronal differentiation-induced hAFSCs. RT-qPCR data are represented as relative fold change over 2D undifferentiated control, normalized for the housekeeping genes GAPDH and RPL13A; values are indicated as mean ± SD (n = 3). Note: * denotes significant difference with *p* < 0.05, ** denotes significant difference with *p* < 0.01, as evaluated using one-way repeated-measures ANOVA with Tukey‘s post hoc test.

**Figure 4 ijms-24-03584-f004:**
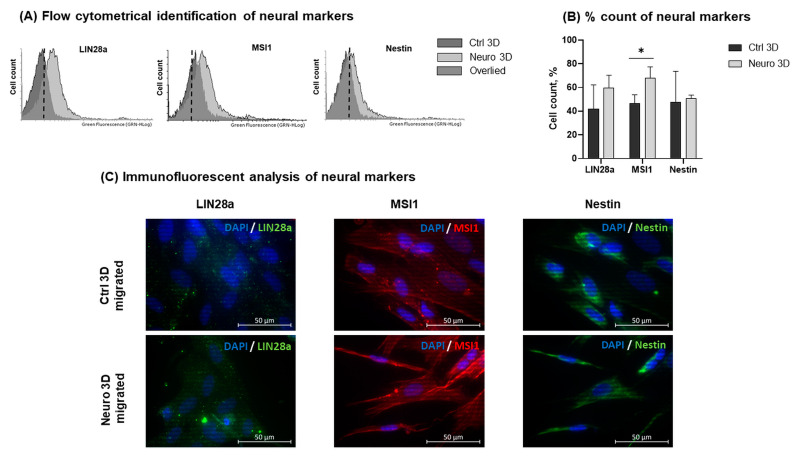
Protein expression profile in control hAFSCs and hAFSCs induced to neural differentiation, when cells were migrated from 3D spheroids. (**A**,**B**) Expression analysis of neuronal differentiation associated proteins Nestin, MSI1 and LIN28a in control and neuro-induced hAFSCs migrated from 3D spheroids. Neurogenic differentiation was initiated using chemical cocktail consisting of 50 ng/mL BDNF + 100 ng/mL NGF + 5 mM KCl + 2 μM RA. For spheroid formation, 10,000 hAFSCs were seeded to Nunclon Sphera 96-well plates. Spheroids were cultivated for 7 days. For hAFSCs migration from 3D cultures, spheroids, after 7 days of cultivation in Nunclon Sphera 96-well plates, were transferred to standard cell culture dishes and were left for 5 days to migrate. Protein expression was evaluated flow cytometrically, estimating the percent of positive cells. (**A**) Representative flow cytometry images and (**B**) mean values ± SD (n = 3). Note: * denotes significant difference with *p* < 0.05, as evaluated using Student‘s *t*-test. (**C**) Immunofluorescence analysis of Nestin, MSI1 and LIN28a expression in control and neuro-induced hAFSCs migrated from 3D spheroids. Representative images of control (Ctrl 3D migrated) and neuro-induced (Neuro 3D migrated) hAFSCs, showing positive cells for LIN28a (green), MSI1 (red) and Nestin (green). Nuclei were counterstained with DAPI (blue); scale bar = 50 µm.

**Figure 5 ijms-24-03584-f005:**
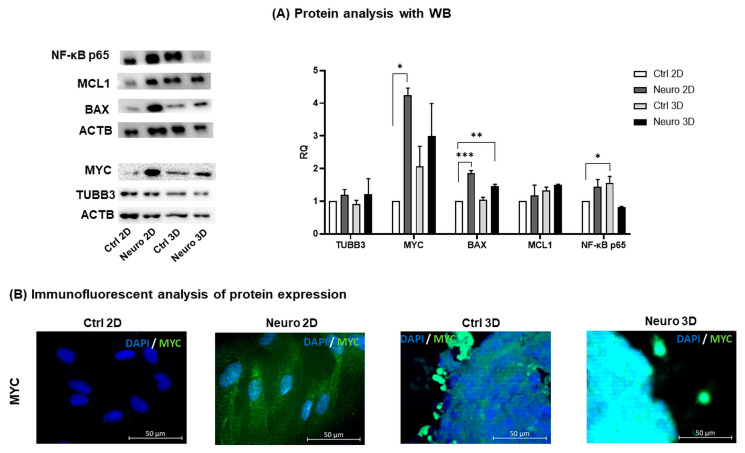
Protein expression profile in control hAFSCs and hAFSCs induced to neural differentiation, when cells were grown in 2D and 3D cultures. (**A**) Representative Western blots of proteins from control (untreated) and neuro-differentiated hAFSCs, grown in 2D and 3D cell cultures. Loading control ACTB was used for normalization of each protein band. Numerical values in the graph denote relative fold change over 2D undifferentiated control, values are indicated as mean ± SD (n = 3). Note: * denotes significant difference with *p* < 0.05, ** denotes significant difference with *p* < 0.01, *** denotes significant difference with *p* < 0.005, as evaluated using one-way repeated-measures ANOVA with Tukey‘s post hoc test. (**B**) Immunofluorescence analysis of MYC in control and neuro-induced hAFSCs, grown in 2D and 3D cultures. Representative images of control 2D (Ctrl 2D), neuro-induced 2D (Neuro 2D), control 3D (Ctrl 3D) and neuro-induced 3D (Neuro 3D) hAFSCs cultures, showing positive cells for MYC (green). Nuclei were counterstained with DAPI (blue); scale bar = 50 µm.

**Figure 6 ijms-24-03584-f006:**
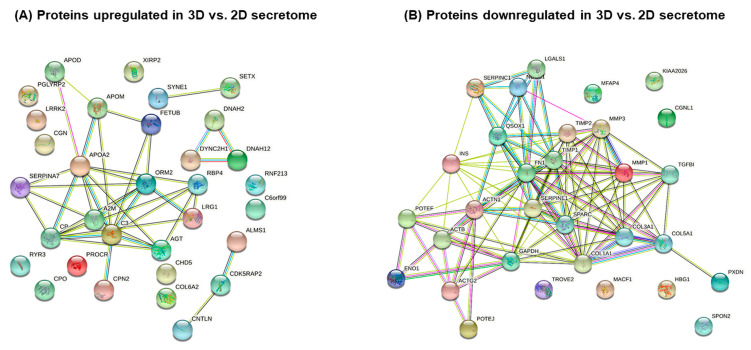
Proteins identified in hAFSCs secretome. (**A**) Identified proteins that were upregulated in 3D vs. 2D hAFSCs secretome. (**B**) Identified proteins that were downregulated in 3D vs. 2D hAFSCs secretome. Association networks of proteins were studied and represented using STRING database (http://string.embl.de, accessed on 13 July 2021).

**Table 1 ijms-24-03584-t001:** The 10 most upregulated and downregulated proteins in 3D hAFSCS secretome sorted by the fold change values.

	Upregulated in 3D Secretome		Downregulated in 3D Secretome	
No.	Protein Name	Fold Change	Protein Name	Fold Change
1	Alpha-2-macroglobulin (A2M)	11–18	Plasminogen activator inhibitor 1 (SERPINE1)	25–484
2	Complement C3 (C3)	5–15	Transforming growth factor-beta-induced protein (TGFBI)	82–409
3	Ryanodine receptor 3 (RYR3)	4–14	Secreted protein acidic and rich in cysteine (SPARC)	49–195
4	Leucine-rich repeat serine/threonine-protein kinase 2 (LRRK2)	4–10	Fibronectin (FN1)	72–83
5	Dynein axonemal heavy chain 2 (DNAH2)	2–5	Metalloproteinase inhibitor 1 (TIMP1)	36–83
6	Immunoglobulin kappa constant (IGKC)	3	Collagen alpha-1(V) chain (COL5A1)	35–36
7	E3 ubiquitin-protein ligase RNF213 (RNF213)	1.5–3.2	Collagen alpha-1(I) chain (COL1A1)	20–28
8	Centlein (CNTLN)	1.9–2.2	Galectin-1 (LGALS1)	8–17
9	Putative uncharacterized protein LINC02901 (C6orf99)	1.8–2.2	Nucleobindin-1 (NUCB1)	4–8
10	Dynein axonemal heavy chain 12 (DNAH12)	1.6–2.2	Glyceraldehyde-3-phosphate dehydrogenase (GAPDH)	7–9

## Data Availability

The datasets (data in native formats) used to support the findings of this study are available from the corresponding author upon request.

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
