# Peer review of "Effect of 3D Spheroid Culturing on NF-κB Signaling Pathway and Neurogenic Potential in Human Amniotic Fluid Stem Cells"

_ijms, 2023, doi:10.3390/ijms24043584_

Round 1
Reviewer 1 Report
The aim of the study described in this manuscript is clear: is 3D culturing of hASFSCs for regenerative medical purposes more attractive than 2D culturing?
However, the results of the study and the conclusions are are less clearly described. I will try to describe comprehensively where improvement in the presentation of the study is needed:
1. Abstract. It is not clear why hAFSCs are attractive (In Introduction it is stated that these cells are needed for regenerative medicine (line 35). This should be mentioned in Abstract and it should be discussed more in the Introduction.
2. I agree that harvesting single hAFSCs in amniotic fluid is more easy than that of MSCs from tissues, but then it should be explained why it is expected that 3D culture of these single cells would be more attractive. Both in Abstract and Introduction.
3. In Abstract but also in Results, differences between 2D and 3D in expression of speciofic genes are not compared. See e.g. lines 15-17 and Figure 1. Fig. 1A is only 2D and Fig. 1B does not show differences. Figures 2 and 3 are extremely difficult to interpret also because the legends are very complex and lengthy.
4. The relevance of Table 1 is not clear to me.
5. Ther conclusions in Abstract (lines 22-25) and in the Conclusions section (lines 528-535) are very general and hardly based on the results.
6. The text of the Results section and the Discussion section is also extremely difficult to understand because of the very detailed presentions.
Reviewer 2 Report
There is a long ongoing discussion whether the conventional two-dimensional culturing of cells is the appropriate way for biomedical research as cells grow in the body in three dimensions and it has been shown in the past that cells cultured in two dimensions differ in their gene expression profile from cells cultured in three dimenstions.
Giedre Valiulienè and colleagues have now extended this discussion to amniotic fluid stem cells. They have cultured these cells in the conventional two-dimensional way or in three dimensions and compared the expression of several genes, particularly those related to stemness and to the NF-kB pathway.
As shown before for MSCs from adipose tissue, amniotic fluid stem cells show higher expression of stem cell markers such as sox2, oct4 and nanog when they are cultured in three dimensions. Also some other genes like msi1 and msi2 or nse and ntrk1 also showed higher expression in the three-dimensional culture while others like tp53 were not affected or even downregulated. The same was true for microRNAs.
Although it is a truly descriptive paper, the results are important for the field as it shows that culture conditions are extremely important for cell behaviour and also that a change in culture conditions can improve the stemness characteristics of stem cells and as such eventually prolong their culture time.
Minor comment:
Fig. 1B: untreated and neuro-induced cells look very similar. Maybe there is a better way to show their differences than a microscopic picture (e.g. analysis of gene expression etc.)
Reviewer 3 Report
The manuscript submitted by ValiulienÄ—et al. demonstrated that cultivation of hAFSCs in 3D spheroids increases expression of stemness markers and modulates NF-κB signaling pathway. The study needs some improvements in order to be suitable to be published in this journal..
Major concerns
Fig 4c – IF images of MSI1 staining are not convincing since this marker should have a nuclear localization.
Fig 5a – WB should be replaced with one obtained loading a similar (even if not identical) amount of proteins, sine the evident differences in the actin levels make it difficult to consider these lanes as a control of loading and thus to normalize the densitometry to these values
Fig 5b- IF images of MYC staining are not convincing since this marker should have a nuclear localization.
Round 2
Reviewer 1 Report
The authors have revised the manuscript thoroughly and, in my opinion, it is now acceptable for publication
Author Response
We would like to thank Reviewer for his/her comments and suggestions as it helped to improve our manuscript.
Reviewer 3 Report
Dear Authors,
the answers you have produced on the immunofluorescence images are satisfactory, while those concerning the WB are not. Within the same gel the samples should be comparable in charge but both beta-actins shown are very different. The authors do a densitomtric analysis and argue that there are differences between the samples, but they do not show a statistical analysis. I reiterate that the samples should be reloaded and densitometry re-evaluated.
Author Response
Regarding Reviewer’s concern, we have reloaded samples of BAX, MCL1 and NF-κB p65 (in this case, control ACTB protein showed more equal distribution compared to the previous data). However, we were unable to get notably more equal distribution in ACTB protein blots when testing for TUBB3 and MYC protein expression. Nevertheless, additional data obtained from densitometry was added for statistical analysis. We have to admit that indeed in the previous version of manuscript statistical analysis for WB data was lacking. Therefore, in the revised version of manuscript statistical analysis was performed and significance of data was denoted (Figure 5A, lines: 377-381). We would like to thank Reviewer for this remark.
Round 3
Reviewer 3 Report
The paper is now suitable for publication, in my opinion.